# LoBaSS: Gauging Learnability in Supervised Fine-tuning Data

## Abstract

Supervised Fine-Tuning (SFT) serves as a crucial phase in aligning Large Language Models (LLMs) to specific task prerequisites. The selection of fine-tuning data profoundly influences the model's performance, a choice traditionally grounded in data quality and distribution. In this paper, we introduce a new dimension in SFT data selection: learnability. This new dimension is motivated by the intuition that SFT unlocks capabilities acquired by a LLM during the pretraining phase. Given that different pretrained models have disparate capabilities, the SFT data appropriate for one may not suit another. Thus, we introduce the term "learnability" to define the suitability of data for effective learning by the model. We present the **Lo**ss **Ba**sed **S**FT Data **S**election (LoBaSS) method, utilizing data learnability as the principal criterion for the selection SFT data. This method provides a nuanced approach, allowing the alignment of data selection with inherent model capabilities, ensuring optimal compatibility and learning efficiency. In experimental comparisons involving 7B and 13B models, our LoBaSS method is able to surpass full-data fine-tuning at merely 6% of the total training data. When employing 16.7% of the data, LoBaSS harmonizes the model's capabilities across conversational and mathematical domains, proving its efficacy and adaptability.

## 1 Introduction

Large Language Models (LLMs) (Brown et al., 2020; Chowdhery et al., 2022; Touvron et al., 2023; Ouyang et al., 2022) have sparked a revolution in the field of Natural Language Processing (NLP), with far reaching impacts in domains such as law (Cui et al., 2023), medical (Singhal et al., 2022) and finance (Wu et al., 2023). A critical step in aligning LLMs to human preference is Supervised Fine-tuning (SFT), which enables pretrained models to exhibit strong instruction-following capabilities (Chung et al., 2022; Ouyang et al., 2022; Touvron et al., 2023; Wang et al., 2022; Zheng et al., 2023). While the selection of training data is important for all stages of LLM training, it is particularly important for the SFT stage where a few thousand of carefully curated data enables finetuned model to demonstrate remarkable performanceZhou et al. (2023).

In general, there have been two primary approaches to obtaining fine-tuning data: 1) distilling data from powerful teacher models (Taori et al., 2023; Xu et al., 2023) , and 2) using manually annotated data (Zhou et al., 2023). In determining what constitutes good fine-tuning data, a common consensus is that valuable data is of high quality and diversity (Ji et al., 2023; Zhou et al., 2023; Chen et al., 2023b;a). In particular, it is commonly assumed that the quality of the data ensures that the fine-tuned model learns accurate and reliable information, while its diversity helps the model generalize better to a wide range of tasks and scenarios. In practice, for example, Alpagasus (Chen et al., 2023b)contends that low-quality data within the dataset is detrimental and utilizes GPT-4 (OpenAI, 2023) to assess data quality and select for higher-quality data. Humpback (Li et al., 2023a) also employs a powerful language model for data selection, while concurrently conducting iterative rounds of fine-tuning and filtering. On the other hand, works such as (Chen et al., 2023a) rely on sampling from clusters of prompt embeddings to preserve data distribution.

However, despite the progress made in previous works on SFT data selection, we argue that these methods do not take into account the model's intrinsic capabilities and what data will best suited for a given model. As argued in the "Superficial Alignment Hypothesis" (Zhou et al., 2023), the fine-tuning process unlocks the capabilities of pretrained LLMs, which implies that the selection of

data for fine-tuning should be tightly coupled to the model of choice. In this work, we introduce a new dimension for constructing fine-tuning datasets by proposing the criterion of data *learnability*, where we assert that data with high learnability should meet the following three constraints: **i) Data lacking informative content for the model should be avoided. ii) Data that is excessively demanding for the model should be avoided. iii) Data that can be learned more effectively by the model during the fine-tuning process is preferable.**

To fulfill these constraints, we put forward our loss-based supervised fine-tuning data selection method (LoBaSS). This method calculates the learnability scores of data points by measuring their loss with respect to both the pretrained model and a fine-tuned reference model. Subsequently, we select the data points with the highest scores. In order to evaluate the effectiveness of our proposed approach, we conduct experiments using the 7B and 13B LLaMA models (Touvron et al., 2023) on the Alpaca open source dataset (Taori et al., 2023). We select the Self-Instruct (Wang et al., 2022), Vicuna (Zheng et al., 2023), Koala (Geng et al., 2023), OpenAssistant (Köpf et al., 2023), and Helpful Base (Bai et al., 2022) as test datasets. We employ both manual comparisons and used GPT-4 as the referee for evaluation (Zheng et al., 2023). The comparison includes models fine-tuned with the full dataset and models fine-tuned with data filtered using the CharGPT. Figure 1 illustrates our experimental results, showing that our fine-tuned models, using only 6.15% of the data, significantly outperform models fine-tuned with the full dataset and those fine-tuned with data filtered using the ChatGPT. At the same time, we conduct data mixing experiments to validate the effectiveness of our approach in scenarios involving data blending. We achieve a balance between mathematical and general conversation capabilities using 16.7% of the full dataset

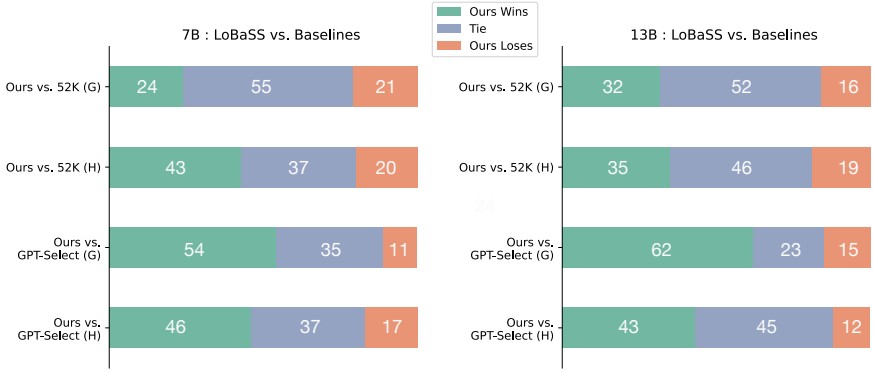

Figure 1: **Our method outperforms full data fine-tuning and ChatGPT filtering method.** The comparison presented the performance of the models fine-tuned with data selected using our method(3,000 items), the full Alpaca dataset and data filtered using ChatGPT (9,229 items). In this context, "G" represents using GPT-4 as the judge, and "H" represents human judgment.

To summarize, our contributions are as follows:

1. Differing from quality and distribution, we propose a novel perspective of evaluating fine-tuning data based on **learnability**, introduce a **quantifiable** metric for the selection of SFT data.

2. We propose the LoBaSS method, which leverages data learnability as the starting point and employs a local model for **secure**, **efficient**, and **high-quality** data selection.

3. In experiments with the 7B and 13B models, we surpassed the performance of the full dataset using **6%** of the data. With **16.7%** of the full dataset, we achieved a balance in the model's capabilities in both conversation and mathematical domains.

## 2 RELATED WORK

**Supervised Fine-tuning.**    In the current alignment process of Large Language Models (LLM), Supervised Fine-tuning (SFT) plays a pivotal role. This step aims to fine-tune the LLM using a small amount of data to enable it to follow human user commands. Self-Instruct (Wang et al.,

2022) generates a significant volume of data for SFT using seed prompts and teacher models. This approach has led to the development of numerous models trained using distillation methods with powerful models (e.g. GPT-4), such as Alpaca (Taori et al., 2023) and WizardLM (Xu et al., 2023).

Apart from distillation with strong models, human-generated data also serves as a high-quality source for SFT data. InstructGPT (Ouyang et al., 2022), for instance, utilizes manually annotated data as a source for SFT in the Reinforcement Learning from Human Feedback (RLHF) method. Vicuna (Zheng et al., 2023), on the other hand, leverages user interaction data to construct the SharedGPT dataset.

**Data for Supervised Fine-tuning.**    In the context of SFT, data's excellence stands as the most pivotal concern, as it directly determines the performance of the fine-tuned model. It is widely acknowledged that the quality of an SFT dataset hinges on two key aspects: firstly, the distribution of the data should ideally be uniform and aligned with the requirements of the intended usage scenarios. Works such as (Xie et al., 2023a; Ji et al., 2023; Chen et al., 2023a) focus on the data distribution to enhance training efficiency. Secondly, data quality is generally deemed more critical than quantity during the SFT process. LIMA (Zhou et al., 2023), for example, suggests that the effectiveness of SFT with a small set of high-quality data significantly surpasses that of large-scale datasets.

In this paper, we introduce a novel perspective on assessing data quality, emphasizing the learnability of the data by the model. This implies that the data should align with the model's current capabilities and offer the potential for greater improvements in performance.

**Data Selection.**    Past methods such as DoReMi (Xie et al., 2023a), DRO (Oren et al., 2019), RHO (Mindermann et al., 2022), and DSIR (Xie et al., 2023b) have primarily focused on data selection during pre-training, and RHO also uses loss based method. However, in the context of SFT, there are significant differences in data distribution compared to pre-training. Additionally, SFT's objective, which is to follow human instructions, is closely tied to model capabilities. Recent SFT data selection approaches, like AlpaGasus (Chen et al., 2023b), employ ChatGPT to assess data quality, which carries the risk of data leakage and considers only the inherent quality of the data. Humpback (Li et al., 2023a) utilizes complex backtranslation processes, whereas our method is comparatively straightforward and efficient. Instruction Mining (Cao et al., 2023) employs multiple indicators for data selection, while our approach leverages a reference model to highlight data learnability.

## 3 METHOD

### 3.1 OVERVIEW

Besides data distribution and data quality, we argue that the data's learnability is a key factor influencing its excellence for a model. Therefore, we propose the LoBaSS method, which starts from the perspective of data learnability, to explore what kind of data can be better learned in the SFT process, to further guide the construction of data sets in the SFT stage, reduce the training cost of SFT, and improve the training effect of SFT.

The main procedure can be divided into two steps: first, using the full data to fine-tune the pretrained model (what we refer to as the initial model here), to obtain the reference model; then, using the reference model and the initial model to calculate the reference loss and the initial loss of each data point, and then obtain the score of each data point through these two losses. Subsequently, sort the scores to obtain the top-ranked data points as the selected dataset. Figure 2 shows the overview of the method.

### 3.2 INITIALIZATION

**Full Dataset.**    The target of LoBaSS is to select an efficient subset from a large SFT dataset. We need a SFT dataset, which we refer to as the full dataset $D_{\text{full}}$. Each data point in the full dataset is formatted as $\{x_i, y_i\}$, where $x_i$ represents the prompt and $y_i$ represents the response to this prompt. The prompt is composed of a consistent prompt template and instructions.

**Initial Model.**    LoBaSS does not use online model services. It uses only local models. We need a local pretrained language model, which we refer to as the initial model $M_{\text{ini}}$. In contrast, the

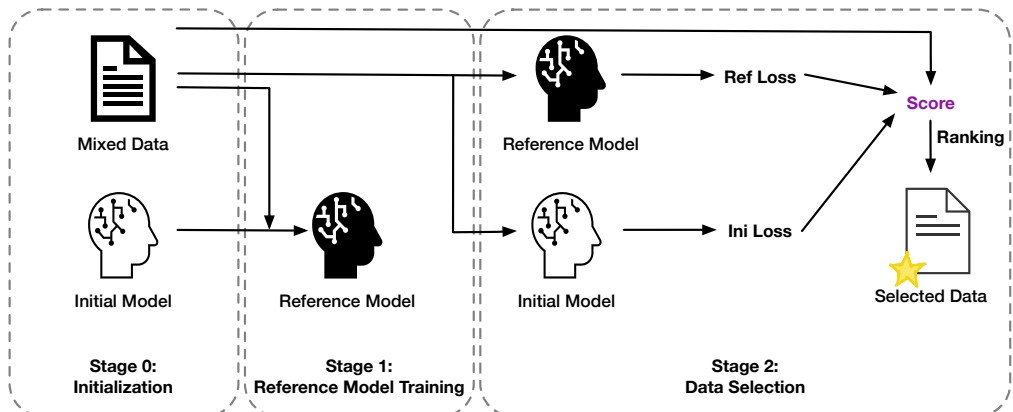

Figure 2: The overview of LoBaSS method. We start from a pretrained model, e.g.LLaMA and mixed SFT dataset, e.g. Alpaca. **Reference Model Training**: the initial model is fine-tuned with the full dataset to get the reference model. **Data Selection**: both loss of reference model and initial model is used to compute the score of each data point and the score is then ranked for selection.

language model obtained by supervised fine-tuning a pretrained model with mixed dataset $D_{\text{ori}}$ is referred to as the reference model $M_{\text{ref}}$.

## 3.3 SELECTION FUNCTION

Previous work on SFT data screening mainly focused on the quality of data and the distribution of data, without considering the specificity of data to models. We argue that the difficulty and value of learning the same data by different models are different. Whether a specific model can learn certain data well is defined by us as the **learnability** of the data in the previous section.

What kind of data has good learnability? We give three constraints: **i) Data lacking informative content for the model should be avoided**, **ii) Data that is excessively demanding for the model should be avoided**, and **iii) Data that can be learned more effectively by the model during the fine-tuning process is preferable**;

We now mark a fine-tuned model $M_{\text{ref}}$ that calculates the SFT loss for a data point $(x_i, y_i)$ through a given loss function as $L_{\text{ref}}(x_i, y_i)$ and the loss of the pre-trained model $M_{\text{ini}}$ for this data point as $L_{\text{ini}}(x_i, y_i)$.The loss function we use in practice is the cross-entropy loss as shown in Equation 1. Next, we will discuss these three constraints in detail to come up with a formula that meets these three constraints we proposed.

$$L(x, y) := \frac{\sum_{y^i \in y} - \log p(y^i|x)}{\text{Len}(y)} \tag{1}$$

**Constraint 1. Data lacking informative content for the model should be avoided.**
When a task can already be effectively performed by a pre-trained model, there is no need to fine-tune the model extensively on this task, and thus, such data holds limited value for model fine-tuning. This type of data lacks informative content for the model, resulting in marginal performance improvements during the fine-tuning process. Therefore, the introduction of such data should be avoided during fine-tuning. We measure the informativeness of a data point $(x_i, y_i)$ by its loss value, determining whether it provides any additional information to the model. If a data point exhibits both low $L_{\text{ini}}(x_i, y_i)$ and $L_{\text{ref}}(x_i, y_i)$, it suggests that $(x_i, y_i)$ lacks informative content for the pre-trained model. To adhere to Constraint 1, such data should be screened out.

**Constraint 2. Data that is excessively demanding for the model should be avoided.**
When a task is challenging both for a pre-trained model and for the model after fine-tuning, it is excessively demanding for the model, meaning that the task's difficulty surpasses the model's capability. When a piece of data is incomprehensible or overly challenging for the model, introducing such data during fine-tuning is detrimental. Therefore, we should also avoid introducing such data during the fine-tuning process. We similarly measure the data point's difficulty for the model by

examining its loss. If the $L_{\text{ini}}(x_i, y_i)$ and $L_{\text{ref}}(x_i, y_i)$ of a data point $(x_i, y_i)$ are both high, then it indicates that $(x_i, y_i)$ is is excessively demanding for the model. To adhere to Constraint 2, such data should also be screened out.

**Constraint 3. Data that can be learned more effectively by the model during the fine-tuning process is preferable.**

When a task is challenging for a pre-trained model but the model can complete this task after fine-tuning, we consider that the data has been efficiently learned by the model. When a data point has been efficiently learned by the model, it indicates that this data point holds meaningful learning significance during the fine-tuning process. We can observe whether a data point is effectively learned by the model by comparing the difference between the loss before and after fine-tuning. Specifically, for a data point $(x_i, y_i)$, if $L_{\text{ini}}(x_i, y_i)$ is high, it indicates that the pretrained model cannot perform well on this task; if $L_{\text{ref}}(x_i, y_i)$ is low, it indicates that after fine-tuning, the model has learned its information well and can complete this task. To adhere to Constraint 3, such data points are considered to be selected and retained.

Considering the three constraints described above, we need to remove data points $(x_i, y_i)$ with both $L_{\text{ref}}(x_i, y_i)$ and $L_{\text{ini}}(x_i, y_i)$ too small or too large, and retain data points with $L_{\text{ini}}(x_i, y_i)$ large and $L_{\text{ref}}(x_i, y_i)$ small. These are the three principles we use to select data. Based on these principles, we propose Equation 2 to score different data points.

$$S(x_i, y_i) = L_{\text{ini}}(x_i, y_i) - L_{\text{ref}}(x_i, y_i) \tag{2}$$

Then, we need to select the Top-K scoring data points, which can be expressed as Equation 3. When $S(x_i, y_i)$ is large, it means that the difference between $L_{\text{ini}}(x_i, y_i)$ and $L_{\text{ref}}(x_i, y_i)$ is large. Let's verify whether this equation can meet the three constraints proposed: if $L_{\text{ini}}(x_i, y_i)$ is small and $L_{\text{ref}}(x_i, y_i)$ is small, then $S(x_i, y_i)$ should be small, and $(x_i, y_i)$ will be screened out, meeting Constraint 1; if $L_{\text{ini}}(x_i, y_i)$ is large and $L_{\text{ref}}(x_i, y_i)$ is large, then $S(x_i, y_i)$ will also be small and $(x_i, y_i)$ will be screened out, meeting Constraint 2; if $L_{\text{ini}}(x_i, y_i)$ is large and $L_{\text{ref}}(x_i, y_i)$ is small, then $S(x_i, y_i)$ will be relatively large and $(x_i, y_i)$ will be selected, meeting Constraint 3. Therefore, Equation 3 can well meet the three constraints we proposed.

$$D_{\text{select}} = \text{topk}_{(x_i, y_i) \in D_{\text{ori}}} \left( L_{\text{ini}}(x_i, y_i) - L_{\text{ref}}(x_i, y_i) \right) \tag{3}$$

### 3.4 NORMALIZATION

The equation proposed in the previous subsection may have a potential problem: if $L_{\text{ini}}(x_i, y_i)$ corresponding to $(x_i, y_i)$ is particularly large and $L_{\text{ref}}(x_i, y_i)$ is also large, it is possible to meet the requirement of a large $S(x_i, y_i)$, but such data clearly does not meet our expectations. We observe the presence of this issue in our experiments, which is elaborated upon in detail in Appendix A.1.

To solve this problem, we introduce a normalization term into the formula for score calculation. This introduces a question of choosing $L_{\text{ref}}$ or $L_{\text{ini}}$ as the normalization term. In fact, this choice will not change the order of ranking. We prove this in Appendix A.2. In this article, we choose $L_{\text{ini}}$ as the normalization term, so we can obtain the Equation 4 for calculating the score and the Equation 5 for data selection.

$$S_{\text{norm}}(x_i, y_i) = \frac{L_{\text{ini}}(x_i, y_i) - L_{\text{ref}}(x_i, y_i)}{L_{\text{ini}}(x_i, y_i)} \tag{4}$$

$$D_{select_{\text{norm}}} = \text{topk}_{(x_i, y_i) \in D_{\text{ori}}} \left( \frac{L_{\text{ini}}(x_i, y_i) - L_{\text{ref}}(x_i, y_i)}{L_{\text{ini}}(x_i, y_i)} \right) \tag{5}$$

## 4 EXPERIMENTS

### 4.1 EXPERIMENTAL SETUP

**SFT Dataset.** To explore the effectiveness of this method on both high-quality and low-quality data, we conduct experiments using high-quality and low-quality datasets respectively. The prompts

of the high-quality and low-quality datasets are identical and both come from the Alpaca dataset. The responses of the high-quality data are generated by the GPT-4, which we call **Alpaca-4**, while the responses of the low-quality data are generated by the GPT-3.5-Turbo, which we call **Alpaca-3.5**. Both datasets contain 52,002 English content and conform to our definition of Full Dataset in the previous section.

**Backbones and Baselines.** To explore the scalability of this method, we select 7B and 13B LLaMA (Touvron et al., 2023) models as our backbones. We choose Text-Davinci-003 (Ouyang et al., 2022) as our baseline model. This model, based on GPT-3, has been trained using the RLHF technique. It has undergone several stages, including initial training with manually labeled data for SFT and subsequent fine-tuning using the Proximal Policy Optimization (PPO) (Schulman et al., 2017) algorithm. Text-Davinci-003 exhibits strong adherence to user instructions and demonstrates proficient performance.

We select the methods of random sampling and ChatGPT-based data filtering (Chen et al., 2023b) as our baseline approaches for comparison, thus demonstrating the effectiveness and superiority of our method. Using ChatGPT for data filtering is a widely adopted method for supervised fine-tuning (SFT) data selection. In this approach, ChatGPT assigns a quality score to each data point in the dataset, ranging from 1 to 5 as an integer, and then filters out the data points with high-quality scores to create the selected dataset.

**Test Dataset.** We used a mixed dataset as our test set, including the HH-RLHF, Koala, Self-Instruct, Open Assistant, and Vicuna datasets. The test set consists of 800 test data, each of which is a prompt, covering multiple aspects of daily use, such as generating, math, and coding, and can test the model's ability to follow instructions.

**Evaluation Method.** We use two evaluation methods in this paper. One is the Fastchat (Zheng et al., 2023) method, used to compare the models trained with the full data and the models trained with the filtered data to obtain the relative results. The second method is the AlpacaEval (Li et al., 2023b)method, used to compare the models trained with the filtered data with the fixed baseline model (e.g. Text-Davinci-003) to obtain the absolute results. We employ the GPT-4 as the judge and we also employ human annotators for judging.

## 4.2 SCALING ANALYSIS

To verify the effectiveness of this method on high-quality and low-quality datasets, we selected Alpaca-3.5 and Alpaca-4 for experimentation; at the same time, to verify the effectiveness of this method on models of different sizes, we selected 7B and 13B models for experimentation. From the experimental results, it can be observed that the LoBaSS method achieves superior results compared to fine-tuning with the full dataset, even when using only around 6% of the data, whether it is a high-quality or low-quality dataset. We also conduct a detailed analysis of the selected data in Appendix A.3.

**Our Method vs. ChatGPT Selecting** We compare the performance of models fine-tuned based on data selected by our method with models fine-tuned on ChatGPT selected data to assess the effectiveness of our approach in comparison to ChatGPT selecting. Figure 1 illustrates the results of this experiment.

Since the quality scores generated by the ChatGPT selecting method are discrete integers, we are only able to select a specific number of data points for training. In this experiment, we choose 9,229 data points as a subset and fine-tuned the backbones on this dataset to establish the baseline. Similarly, we use our method to select 3,000 data points as a subset and fine-tune the backbones on this dataset. Another baseline involves fine-tuning the backbones with the entire dataset. The training parameters in the experiment are kept identical.

We use GPT-4 and human evaluators as referees. During GPT-4 assessments, we perform position swapping to eliminate positional bias. In human evaluations, we do not disclose which model generated each response, and the placement order is randomized to eliminate potential biases. Due to the high cost of human evaluation, our test dataset was randomly selected by taking 20 questions from each of the five datasets mentioned earlier, resulting in a mixed test set of 100 questions in total.

The experimental results indicate that, on the test dataset, models fine-tuned with data selected using our method consistently outperform those fine-tuned with data filtered by ChatGPT and models fine-tuned with the full dataset, whether evaluated through human ratings or GPT-4 based judging.

**Alpaca-3.5-Selected vs. Alpaca-3.5-52k**    We compared models trained with various sizes of datasets filtered by our method and with the full dataset, using the same hyperparameters and using the Vicuna dataset as the test set. To observe the phenomenon more clearly, we define Win Score := $\frac{N_{\text{win}} - N_{\text{lose}}}{N_{\text{total}}} + 1$.

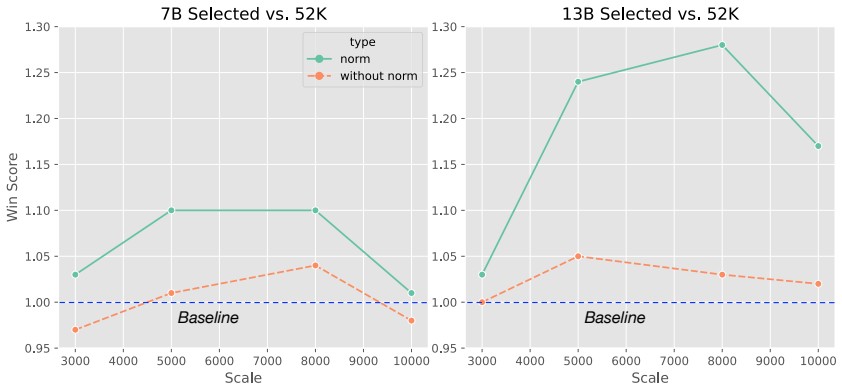

Figure 3: **Models fine-tuned with data selected by LoBaSS surpass the full dataset on Alpaca3.5.** This figure shows the win score comparison between models trained with different sizes of datasets and the full dataset, as well as the improvement brought by using the normalization method. We select the model fine-tuned on the full dataset as the baseline.

From the experimental results in Figure 3, we can see that our method can achieve good results on both 7B and 13B models. With as few as 3,000 data points (**5.77 %**), it can achieve better results than those achieved with the full dataset (52K). We can also observe that after using the improved normalization method, the data filtered by our method is significantly better in quality and the model performance is significantly improved. With our method, less than 6% of the data is required to obtain a model performance exceeding that of a model trained with the full dataset, indicating that in the SFT process, much of the data in the dataset does not contribute significantly to the model fine-tuning or even harms the model performance. We started from the learnability of the data and removed data that does not contribute significantly to the model fine-tuning or is even harmful through data filtering, thereby improving the efficiency and performance of model training.

**Alpaca-4-Selected vs. Text-Davinci-003**    In the Alpaca-3.5 experiment, our method achieved good results, but due to the quality issues of the Alpaca-3.5 dataset, the difficulty of dataset filtering was relatively low. To further validate the versatility of our method, we conducted experiments on the higher-quality Alpaca-4 dataset using 7B and 13B models.

We compare the models fine-tuned with various sizes of datasets filtered by our method with Text-Davinci-003, using a mixed dataset of 800 data points as the test set and using GPT-4 as the judge, with a temperature setting of 0. Since the advantage of the normalization method was already verified in the Alpaca-3.5 experiment, we adopted the normalization method in all experiments on Alpaca-4.

From the experimental results in Figure 4, we can see that our data filtering method still achieve significant results on the high-quality Alpaca-4 dataset. With as few as 800 data points, similar results to those achieved with full-dataset fine-tuning can be achieved. At 3,200 (6.15%) data points, higher results than those achieved with full-dataset fine-tuning can be achieved on both the 7B and 13B experimental conditions. Through comparison with randomly selected data, we can prove that the improvement in model performance is not due to the decrease in the number of data points but rather that our data filtering method effectively selects data that is more learnable and valuable for fine-tuning.

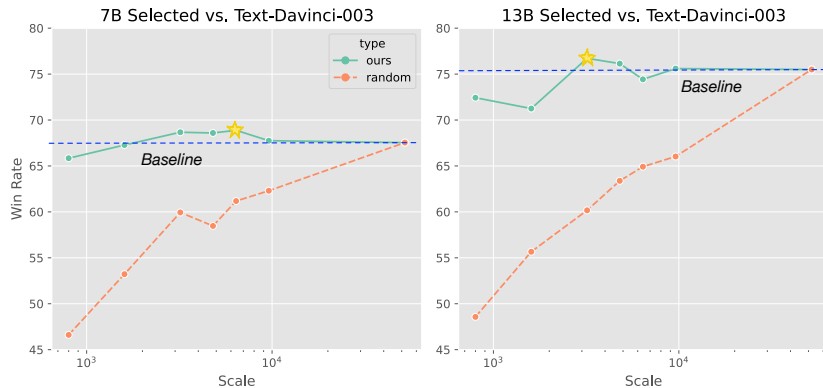

Figure 4: **Models fine-tuned with data selected by LoBaSS surpass the full dataset on Alpaca4.** This figure shows the win rate comparison between models fine-tuned with different sizes of selected datasets and Text-Davinci-003, as well as a comparison with randomly selected data using the same hyperparameters to demonstrate the significant improvement over random data selection. To facilitate comparison, we used a logarithmic horizontal axis. We choose the model fine-tuned on the full dataset as the baseline.

By increasing the size of the selected dataset, we also explore the effect of fine-tuning models with a similar performance to that of the full-dataset model at around 3,000 (6%) data points. After 10,000 (20%) data points, the performance of the fine-tuned models begins to decline, indicating that the fine-tuning performance of the models has saturated within this range. Since the patterns of the 7B and 13B models are generally consistent, we believe that the saturation phenomenon occurs due to the limit on the number of highly learnable data in the dataset rather than the saturation of the model capacity.

## 4.3 DATA MIXING

As introduced in the previous subsection, starting from the perspective of data learnability, our data selection method is capable of effectively compressing LLM fine-tuning data to approximately 6% of the original volume, while achieving results similar to or even better than fine-tuning on the full dataset. In practical applications, one highly meaningful use case for this method is in the context of data mixing.

One significant challenge when fine-tuning large language models is the issue of data mixing. Currently, there exists an imbalance in the quantity of data available from different domains. For instance, there is a plethora of data for general question answering, while acquiring data for mathematical domains can be considerably more challenging. This data imbalance results in an imbalance in the fine-tuned model's capabilities, making data balance a critical factor in fine-tuning large language models.

Our method can be employed for data compression, enabling the reduction of large-scale datasets to smaller ones, which can then be mixed with smaller datasets to balance the multifaceted capabilities of the model. We conducted experiments using the Alpaca-4 dataset to represent easily accessible general question answering data and the GSM8K dataset to represent challenging-to-obtain math and reasoning datasets. We selected Alpaca-4 datasets of varying sizes using our method and combined them with the full GSM8K dataset (7K) to create a blended dataset for fine-tuning the model.

Referring to Figure 5, we can see that when we combine the GSM8K training set with a carefully selected subset from Alpaca-4 and fine-tune the model using this smaller dataset, it effectively balances the model's ability in general tasks and mathematical reasoning. To be more precise, when we add 3200 filtered data points from Alpaca-4 into the mix, we achieve performance on the GSM8K dataset that's similar to fine-tuning solely on GSM8K data, all the while maintaining nearly the same level of general task performance. This represents a notable improvement compared to fine-tuning with the entire Alpaca-4 dataset.

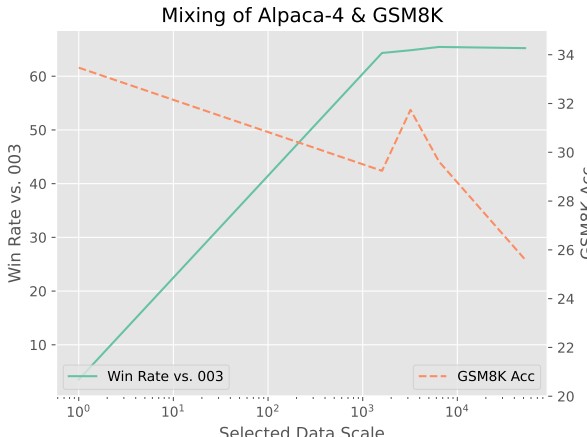

Figure 5: **Our method significantly enhances the effectiveness of data mixing.** The horizontal axis represents the quantity of selected Alpaca-4 data, plotted on a logarithmic scale, while the vertical axis represents GSM8K accuracy and the Win Rate compared to Text-Davinci-003, respectively. In this case, we are using the model of 7B.

We achieved a performance of 98.6% in general tasks and 124.0% in the GSM8K dataset using only 17.2% of the training data compared to the full dataset via our data selection method.

## 5 LIMITATION AND DISCUSSION

One limitation of our work is that while we introduce learnability as a new dimension for measuring SFT data excellence, we primarily focused on methods for only data selection. We did not apply this perspective to the generation and augmentation of SFT data, limiting its potential to enhance model performance. We plan to incorporate the perspective of learnability into the generation and augmentation of data for SFT in the future. This approach will involve creating data that aligns with different model capabilities, further enhancing the effectiveness of fine-tuning.

Furthermore, we do not conduct a specific analysis of how different model capabilities influence the model's selection preferences. Specifically, whether a model's stronger performance in a specific domain directly corresponds to LoBaSS-method-selected data being more inclined toward that domain. We have some preliminary analysis in Appendix A.3, and we plan to delve deeper into this issue in future work.

Another limitation is that in our exploration of data blending and capacity balance, we have not specifically investigated what proportion of data blending would yield better results in terms of capacity balance. This will be an important research direction for our near-term work.

## 6 CONCLUSION

We first introduced learnability as a new perspective to measure the excellence of SFT data, beyond data distribution and quality. We proposed three constraints to define data learnability, and based on these constraints, we introduced a loss-based data selection method for SFT data selection. In our approach, we use the loss of both the backbone and fine-tuned models to calculate the learnability score, and subsequently select the data with the highest scores. Experimental results on the Alpaca dataset demonstrated that fine-tuning with only around 6% of the data can outperform using the full dataset and is also superior to the method of data filtering using GPT-4. Our study offers a novel and effective perspective on how to construct and select datasets for SFT, thereby expanding the understanding for LLMs fine-tuning.

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

# A  APPENDIX

## A.1  NORMALIZATION DETAILS

To verify whether this problem exists, we observe the relationship between the loss and score of selected 6400 data points in $D_{\text{ori}}$ , as shown in Figure 6. We can observe that with high loss values are ranked at the forefront. This may expose the potential problem with Equation 3 we mentioned earlier.

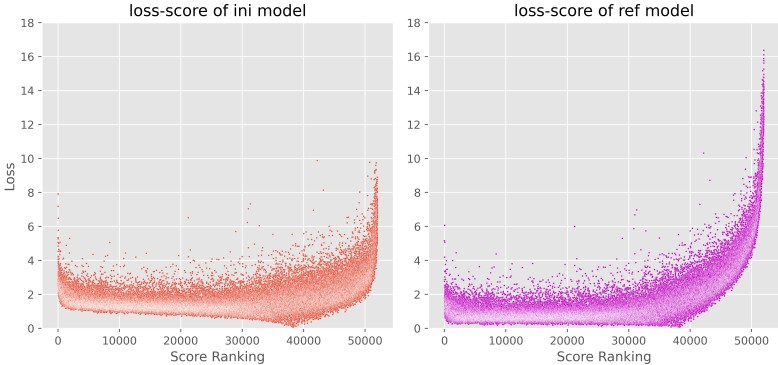

Figure 6: The horizontal axis of this figure is the score ranking, the closer to the left, the smaller the ranking, that is, the larger the score; the vertical axis is the value of the loss. The left figure shows the loss of $M_{\text{ini}}$, and the right figure shows the loss of $M_{\text{ref}}$.

We further analyze the cause of this phenomenon and believe that it may be caused by the calculation method of loss in the fine-tuning process. Because in the fine-tuning process, we mask the tokens corresponding to the prompt for the calculation of loss and only calculate the loss of the tokens corresponding to the response, specifically, we calculate the average cross-entropy loss of each token of the response. Therefore, we believe that the size of the loss may be related to the length of the response. We observe the relationship between score and response length, and the results are shown in Figure 7. At the front of the ranking, there is a phenomenon of increasing length of selected data points. We believe this phenomenon is due to the potential error mentioned earlier.

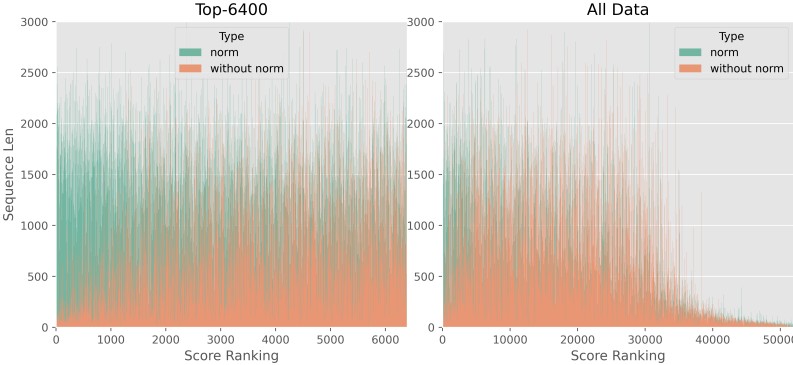

Figure 7: This figure shows the relationship between the length and ranking of the top-6400 subset and 52000 full set before and after using the normalization method. In the top-6400 subset, we can see that the ranking data with higher priority has a shorter length when not normalized, and this problem can be solved by normalization.

After using the normalization method, we observe the relationship between loss and response length again, and the results are shown in Figure 7.

## A.2 PROOF

As long as we can demonstrate that when

$$\frac{L_{\text{ini}}(x_1, y_1) - L_{\text{ref}}(x_1, y_1)}{L_{\text{ini}}(x_1, y_1)} > \frac{L_{\text{ini}}(x_2, y_2) - L_{\text{ref}}(x_2, y_2)}{L_{\text{ini}}(x_2, y_2)}$$

it also satisfies

$$\frac{L_{\text{ini}}(x_1, y_1) - L_{\text{ref}}(x_1, y_1)}{L_{\text{ref}}(x_1, y_1)} > \frac{L_{\text{ini}}(x_2, y_2) - L_{\text{ref}}(x_2, y_2)}{L_{\text{ref}}(x_2, y_2)}$$

then we can achieve our goal.

Let's begin the proof.

When

$$\frac{L_{\text{ini}}(x_1, y_1) - L_{\text{ref}}(x_1, y_1)}{L_{\text{ini}}(x_1, y_1)} > \frac{L_{\text{ini}}(x_2, y_2) - L_{\text{ref}}(x_2, y_2)}{L_{\text{ini}}(x_2, y_2)}$$

we can get

$$1 - \frac{L_{\text{ref}}(x_1, y_1)}{L_{\text{ini}}(x_1, y_1)} > 1 - \frac{L_{\text{ref}}(x_2, y_2)}{L_{\text{ini}}(x_2, y_2)}$$

So

$$\frac{L_{\text{ref}}(x_1, y_1)}{L_{\text{ini}}(x_1, y_1)} < \frac{L_{\text{ref}}(x_2, y_2)}{L_{\text{ini}}(x_2, y_2)}$$

and

$$\frac{L_{\text{ini}}(x_1, y_1)}{L_{\text{ref}}(x_1, y_1)} > \frac{L_{\text{ini}}(x_2, y_2)}{L_{\text{ref}}(x_2, y_2)}$$

Then we subtract one from both sides of the inequality.

$$\frac{L_{\text{ini}}(x_1, y_1) - L_{\text{ref}}(x_1, y_1)}{L_{\text{ref}}(x_1, y_1)} > \frac{L_{\text{ini}}(x_2, y_2) - L_{\text{ref}}(x_2, y_2)}{L_{\text{ref}}(x_2, y_2)}$$

And the proof is complete.

Through the above process, we have successfully demonstrated that choosing either $L_{\text{ref}}(x, y)$ or $L_{\text{ini}}(x, y)$ as the denominator for normalization does not affect the ranking of the learnability score.

## A.3 DATA ANALYSIS

To explore what types of data are required by the model during the SFT process, we conducted a further analysis of the data selected by the 7B and 13B models.

The 7B and 13B models will have different gaps in different abilities (e.g. math, coding, knowledge), so in theory, the data selected by learnability should also be different. Table 1 shows the similarity of the data selected by the 7B and 13B models, which can be seen not to be very high. We used two methods to analyze the differences between the data selected by the 7B and 13B models.

| Scale of Dataset | 800 | 1600 | 3200 | 6400 | 9600 | 26000 | 39000 |
|---|---|---|---|---|---|---|---|
| Num of Same | 516 | 1111 | 2399 | 5123 | 8033 | 24358 | 38075 |
| Same Rate | 0.645 | 0.694 | 0.749 | 0.800 | 0.837 | 0.937 | 0.976 |

Table 1: The number and proportion of data points selected by 7B and 13B models are same. Scale of Dataset refers to the number of top data points after sorting, Num of Same refers to the number of same data points in the two sets, and Same Rate refers to the proportion of the same data points to all the selected data.

**Benepar Method** We used the Benepar method to analyze the distribution of the dataset. The Benepar method divides natural language statements according to the verb predicate and noun object, and hierarchical statistics were carried out with the predicate as the root node and the object as the leaf node.

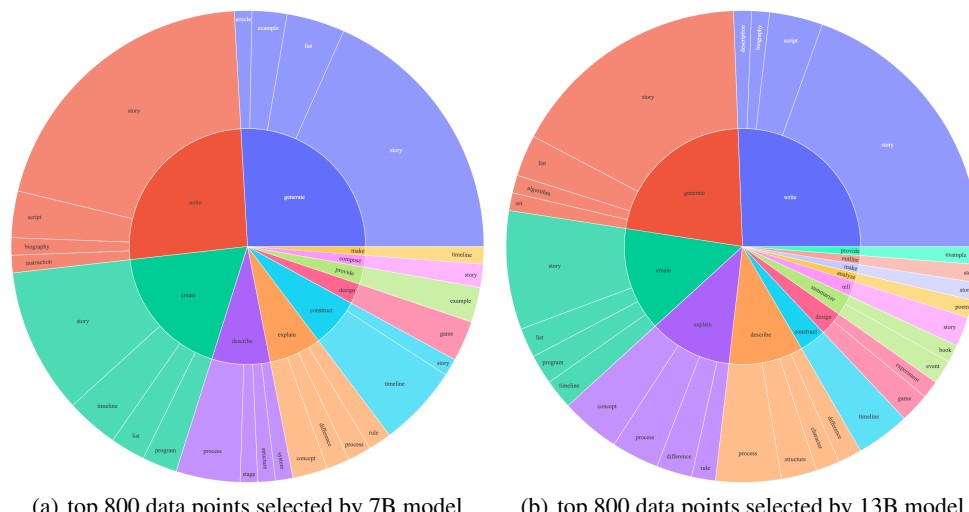

(a) top 800 data points selected by 7B model      (b) top 800 data points selected by 13B model

Figure 8: A comparison of the distribution of data filtered by the 7B and 13B models. The inner ring represents verbs, which are the root nodes for sentence classification, and the outer ring are nouns, which are the leaf nodes for sentence classification. [the text in the figure is too small. -PFLIU]

Figure 8 compares the Benepar distribution of the top 800 data points selected by 7B and 13B models. It can be seen that compared with the 7B model, the data points selected by the 13B model has fewer creation tasks (e.g. write, generate, create) and more interpretative tasks (e.g. explain, describe), and the task types also have better diversity especially in the long tail.

**LLM for classification**      Due to the Benepar method's classification relying on the verb of a sentence as the root node, it may not necessarily effectively convey the meaning of the sentence and thus may not perform well in sentence classification. Considering this limitation, we employ the powerful LLM as a classifier for a more precise classification.

To determine how the data is categorized, we start with Alpaca's seed instructions and used GPT-4 to classify them into seven categories. Subsequently, we use GPT-4 to classify the first 800 data entries selected by models 7B and 13B, as shown in Table 2.

| Category | 7B | 13B | Delta(Rate) |
|---|---|---|---|
| Programming and Coding | 60 | 56 | -4(-6.6%) |
| Planning and Organization | 63 | 57 | -6 (-9.5%) |
| Knowledge and Information Extraction | 275 | 296 | +21 (+7.6%) |
| Language and Text Processing | 53 | 45 | -8 (-15.1%) |
| Creative Writing and Entertainment | 311 | 286 | -25 (-8.0%) |
| Problem Solving and Math | 26 | 46 | +20 (+76.9%) |
| Recommendations and Suggestions | 9 | 8 | -1 (-11.1%) |
| Others | 3 | 6 | – |

Table 2: The number of items in each category selected by 7B and 13B model.We also show the difference between the data selected by 7B and 13B model.

By observing, we can notice that the data in the "Problem Solving and Math" category showed the most significant change between the 7B and 13B data, increasing by 76.9%. This phenomenon partially aligns with our expectations, indicating a substantial difference in mathematical and reasoning capabilities between the 7B and 13B models. Further analysis and investigation are left for future work to complete.

