# OpenReview forum: "Gauging Learnability in Supervised Fine-tuning Data"
_ICLR.cc/2024/Conference — ICLR 2024 Conference Withdrawn Submission_

### Official Review · Reviewer_MMqo · 2023-11-01

**Soundness:** 3 good
**Presentation:** 3 good
**Contribution:** 3 good
**Rating:** 6
**Confidence:** 2

**Summary:**

The authors propose a method for measuring the importance of data used to perform supervised fine-tuning (SFT) of large language models. This measure is used to select the most important data examples in order to select a subset of SFT data that is most beneficial for the fine-tuning process. They introduce a measure of data importance they call "learnability" which is computed with respect to a specific model and expresses three design choices (i.e. constraints) the authors underline, which can be summarized as (1) assign a low score to uninformative data; (2) assign a low score to hard-to-learn data; (3) assign a high score to efficiently learnable data. The authors compare their prioritized selection method against training on all SFT data and against asking Chat-GPT to filter out uninformative data examples. Experimental evaluation demonstrates that their method generally outperforms these two baselines.

**Strengths:**

**S1**: Supervised fine-tuning is becoming a crucial task when applying LLMs to specific scenarios and doing it in a data-efficient way is important.

**S2**: The proposed method is quite simple which makes it easy to apply in practice.

**S3**: The experiments demonstrate the effectiveness of the proposed method and show that it is able to remove some "hurtful" data which ends up giving a better-performing model that was fine-tuned only on a subset of the SFT data.

**Weaknesses:**

**W1**: The problem of selecting optimal subsets of training data is far from novel. It is known as the coreset selection problem and many methods have been proposed. It would be good to add some references to that line of work.

**W2**: The proposed design constraints are argued for using simple reasoning techniques (section 3.3). However, one could also see them as arbitrarily chosen. For example, the statement "When a piece of data is incomprehensible or overly challenging for the model, *introducing such data during fine-tuning is detrimental*." is strange because one might ask where the evidence is that it is detrimental. It would be useful if each of the constraints were empirically shown to be necessary (i.e. individually, not together) using some ablation analysis.

**W3**: The notion of "learnability" is referred to across the paper as a "dimension", "measure" and "perspective". To me, it seems like the best fit would be to call it "measure" and make this uniform across the paper.

**W4**: I would rephrase the x-axis label in Figures 3 and 4 from "Scale" into something more clear (e.g. amount of data selected).

**W5**: There are several strange statements in the related work section: (1) "the distribution of the data should ideally be *uniform* and aligned with the requirements of the intended usage scenarios" -- how can a data distribution be uniform?; (2) "ChatGPT to assess data quality, which carries the risk of data leakage and considers only the inherent quality of the data" -- what is the meaning of the term inherent quality of data?

**Questions:**

**Q1**: Doesn't the introduction of normalization as described in Section 3.4 contradict constraint 2? Namely, constraint 2 implies that we should be able to detect data that is excessively demanding by observing large $L_{ini}$ and $L_{ref}$ values. On the other hand, in section 3.4 it is implied that this data does not "meet our expectations". (also, it is not clear, which expectations?)

**Q2**: What is the difference between the backbone and the baseline model?

**Q3**: What is data mixing (as mentioned in section 3.4)?

---

> ### Author Response · Authors · 2023-11-22
>
> We thank the reviewer for the comment. We plan on extensively revising our manuscript with more extensive benchmarking and will be withdrawing our submission as a result.
> However, we'd like to address a few points raised by the reviewer for clarification.
>
> W1: It would be good to add some references to that line of work.
> Response 1: Thank you for your suggestions. We will incorporate more relevant references in the subsequent updates of our work.
>
> W2: It would be useful if each of the constraints were empirically shown to be necessary (i.e. individually, not together) using some ablation analysis.
> R2: Thank you for your suggestion. We will incorporate ablation experiments on the three constraints in the subsequent updates of our work to better explore the impact of different constraints on the selection outcomes.
>
> W3 & W4: Some suggestions in detail.
> R3 & 4: Thank you for your suggestions. We will address the two points you mentioned in the subsequent versions.
>
> W5: Some strange statements.
> R5: (1)The use of the word "uniform" may not be entirely appropriate in this context. What we intend to convey is that the distribution across various domains should not exhibit strong biases. For instance, we aim to avoid situations where one domain has more than 50% of the data, while some domains have no data at all.
> (2)The term "inherent quality of data" here refers to considering the data on its own, without taking into account its association with the model—whether the data is suitable for the model to learn from, serving as a point of comparison with our method.
>
> Q1: Doesn't the introduction of normalization as described in Section 3.4 contradict constraint 2?
> A1: The introduction of normalization does not conflict with Constraint 2. In Constraint 2, we emphasize the exclusion of data points where both $L_{\text{ref}}$ and $L_{\text{ini}}$ are relatively large. If both of these values are large, their difference remains relatively small after normalization. In other words, there will be a smaller $\text{Score}, so these points will not be filtered out. By using the normalization condition, we can effectively exclude data points that fit the following description: both $L_{\text{ref}}$ and $L_{\text{ini}}$ are large, $L_{\text{ref}}$ has decreased in absolute value compared to $L_{\text{ini}}$, but the relative decrease is small. Experimental results also demonstrate that such data points are effectively filtered out after applying the normalization method.
>
> Q2: What is the difference between the backbone and the baseline model?
> A2: In this paper, the term "baseline" refers to the model fine-tuned using all available data. We employ our method to filter out a smaller subset of data for fine-tuning the model. We then compare the experiment results obtained from fine-tuning our model with the baseline model.
>
> Q3: What is data mixing (as mentioned in section 3.4)?
> A3: "Data mixing" is an application scenario we identified for our method. For example, if we need a model with conversational skills and strong mathematical abilities, we would introduce both dialogue and mathematical data during fine-tuning. However, there is a large quantity of dialogue data and a smaller quantity of mathematical data. If an excessive amount of dialogue data is used, the improvement in mathematical abilities may not be significant. Therefore, we need to select a portion of dialogue data to mix with mathematical data, balancing the use of these data to fine-tune a model with balanced capabilities in both conversation and mathematics. Section 3.4 describes experiments conducted in this manner.

---

### Official Review · Reviewer_b7hv · 2023-11-01

**Soundness:** 2 fair
**Presentation:** 3 good
**Contribution:** 1 poor
**Rating:** 1
**Confidence:** 4

**Summary:**

This paper proposes a way to select fine-tuning data for downstream LLMs using the normalized difference between the pre-trained model and a fine-tuned model, which this paper calls the "reference" model.

**Strengths:**

The paper is easy to understand. Details are explained when necessary.

**Weaknesses:**

* Typo: in page 2, CharGPT --> ChatGPT
* It is unclear what the novelty of the proposed method is. Moreover, the model is compared to random selection and ChatGPT, with little evidence to support that these are state-of-the-art baselines. Details on the ChatGPT-based filtering are scarce. For example, what prompt is used?
* The paper shows results as "wins", "ties", or "losses", without showing a table of actual scores on the test sets.
* The experimental design for the Alpaca-4 experiments (Section 4.2) is flawed. For instance, the normalized method being better in the Alpaca-3.5 experiments does not indicate that it will also be better in the Alpaca-4 experiment.
* The proposed method requires a new "reference" model to be trained via fine-tuning, which can be prohibitively expensive on top of a downstream LLM.

**Questions:**

None.

---

> ### Author Response · Authors · 2023-11-22
>
> We thank the reviewer for the comment. We plan on extensively revising our manuscript with more extensive benchmarking and will be withdrawing our submission as a result.
> However, we'd like to address a few points raised by the reviewer for clarification.
>
> Q1: It is unclear what the novelty of the proposed method is.
> A1: Currently, among all methods for filtering SFT data, none have taken into account the impact of model capability differences on data selection within the SFT context. Considering that the SFT process aims to stimulate the capabilities of pre-trained models, and models with varying capabilities are expected to favor different types of data, this forms the basis for our proposed method.
>
> Q2: There’s no baseline.& What prompt is used?
> A2: We apologize for any confusion. In this context, the "ChatGPT Selection" method actually refers to the widely recognized AlpaGasus method. The experimental results indicate that our approach significantly outperforms AlpaGasus. We conducted an extensive review of recent relevant literature, and AlpaGasus emerged as one of the most effective methods, serving as our chosen baseline. In the "Related Works" section, we also mentioned the Humpback study. However, due to its relatively complex methodology and process, which does not align with a general SFT data selection approach, we did not include it in our comparative analysis. The AlpaGasus dataset is open source, and as such, the prompts used for filtering are derived from AlpaGasus. However, the specific details of these prompts have not been disclosed by the dataset creators.
>
> Q3: The paper shows results as "wins", "ties", or "losses", without showing a table of actual scores on the test sets.
> A3: Currently, most similar tasks involving the scoring of generative tasks by humans and GPT often introduce similar methods, namely comparing "wins," "ties," or "losses" because the comparison is between two models. For instance, leaderboards in projects like Vicuna and AlpacaEval calculate actual scores through ELO scores or by comparing against a fixed reference (such as Text-Davinci-003). Recognizing the diversity of evaluation methods, we employed a "win, tie, lose" relative comparison approach in experiments related to Alpaca-3.5. In experiments related to Alpaca-4, we adopted a method of comparing with Text-Davinci-003 to obtain the actual score. This setup can be considered an advantage in the experimental aspect of our work.
>
> Q4: The experimental design for the Alpaca-4 experiments (Section 4.2) is flawed.
> A4: We argue that the performance improvement introduced by normalization is a universal principle, meaning that normalization can mitigate the loss bias introduced by varying lengths and is independent of the dataset used. We have validated this conclusion on the Alpaca-3.5 dataset, and there is no need to conduct further experiments on the Alpaca-4 dataset. Moreover, the difference between the Alpaca-3.5 and Alpaca-4 datasets lies in the quality of the data, which does not impact the applicability of this conclusion.
>
> Q5:The proposed method requires a new "reference" model to be trained via fine-tuning, which can be prohibitively expensive on top of a downstream LLM.
> A5:We have presented two feasible application scenarios for our approach:
> 1. **Model Iteration:**
>    In the industry, models often require periodic updates and iterations. Since it's not possible to continue fine-tuning, aggregating data and starting the SFT process from the pre-trained model is necessary with each update. In this scenario, our method can be employed for data filtering using older models, thereby improving the utilization efficiency of computational resources.
> 2. **Data Mixing:**
>    When dealing with models from various domains and datasets with significant differences in scale, achieving a balance in their capabilities is essential. Our method can be utilized for data filtering, enabling the selection of a more closely matched quantity of data from different domains. This approach helps achieve a balanced distribution of capabilities across multiple domains for the models.

---

### Official Review · Reviewer_JK8S · 2023-11-02

**Soundness:** 2 fair
**Presentation:** 1 poor
**Contribution:** 2 fair
**Rating:** 3
**Confidence:** 4

**Summary:**

This paper investigates the problem of selecting fine-tuning samples for instruction tuning of pre-trained LLMs. Different from prior work that often targets data quality and distribution, this paper introduces a new aspect for selecting data, dubbed "learnability".

Specifically, a reference model is first obtained by fine-tuning a pre-trained LLM ("initial model") on full instruction tuning samples. Then, 3 constraints are imposed:
1. samples lacking information (samples with a small loss on both reference LLM and initial LLM) are removed;
2. hard samples (samples with a high loss on both LLMs) are removed;
3. "learnable samples" (samples with a high loss on initial LLM but lower loss on reference LLM) are selected.

A set of experiments is conducted on LLaMA-7B/13B and compared with ChatGPT-based filtering methods. The paper claims to achieve better performance after fine-tuning on 3k selected samples compared to fine-tuning on full 52k samples.

**Strengths:**

The perspective of this work is interesting. This angle of sample "learnability" is attractive and feels promising. The problem being investigated is timely.

**Weaknesses:**

- The structure of this paper is rather loose. The writing style is problematic. The paper contains too many non-specific descriptions for the methods of this work ("we focus on the new aspect of learnability") or its results ("we achieve better performance with 6% of data"). Its actual technical contribution or specific methodology is not at all introduced until Section 3.3 on Page 4. The abstract says nothing about what this work actually does and the introduction also remains on the descriptive level. This is not the style for a research paper. Subjective descriptions should be used with discretion and only when necessary. The major technical body of this work should be put to the front in the most straightforward manner. The language throughout the paper needs to stay objective and rigorous.

- The methodology described in Section 3.3 is intuitive but somewhat superficial. I do not mind empirical papers based on insights and intuitions. This is nowadays a major drive for the progression of this field. Yet the description in Section 3.3 is too simple for me to feel comfortable. These criteria for samples being "too hard", "too simple", and "in-between and good" are overly subjective. At least, no analysis is provided to ground it to existing frameworks. I guess this threshold is also set in an ad-hoc manner and needs to be tuned with trial and error in each case. It could provide much higher value if the authors could develop it into a principled framework

- The references in this work lack depth. It focuses overwhelmingly on the work during the past year and does not connect to lines of existing research with a richer history (e.g., learnability of samples, data selection problems, simple or hard samples, etc.).

- The term "supervised fine-tuning" used throughout this work actually refers to instruction-tuning. Supervised fine-tuning has a much broader reference than the case studied in this work, especially when not confined to text-completion LLMs.

- Recently, there is already a wealth of work on this topic of "instruction mining" and I believe many have reported results similar to this work–achieving comparable or better performance with a small fraction of 52k Alpaca instruction samples, which is believed to contain low-quality samples that would hurt the performance. The performance reported in this work isn't particularly stronger than the provided baseline and not many baselines are considered.


- Reproducibility: It is unknown how to set the threshold in the proposed constraints. It seems to be manually picked without a principled method or analytical insights.

- Format: Appendix is not cut from the main paper. The PDF provided for the main paper is this 14-page document.

**Questions:**

Appendix should not be submitted under the main paper.

---

> ### Author Response · Authors · 2023-11-22
>
> We thank the reviewer for the comment. We plan on extensively revising our manuscript with more extensive benchmarking and will be withdrawing our submission as a result.
> However, we'd like to address a few points raised by the reviewer for clarification.
>
> Q1: The structure of this paper is rather loose.
> A1: Thank you for your suggestion. We will consider your feedback in future updates to improve the presentation of the paper.
>
> Q2: The methodology described in Section 3.3 is intuitive but somewhat superficial.
> A2: Thank you for your suggestion. In future work, we will conduct ablation experiments to explore the correlation between constraints and outcomes. We have not experimented to determine the optimal threshold design, which may be part of our future work. Our method produces a ranking, where data ranked higher is believed to better stimulate the capabilities of pre-trained models. You can choose the maximum amount of data for fine-tuning based on your needs.
>
> Q3: The references in this work lack depth.
> A3: We thank the reviewer for the feedback. To clarify, in addition to citing works from the past year, our related works section also references many earlier works in various related domains, including data filtering and pre-training data filtering. However, we do intend to include more references in the next revision.
>
> Q4: The performance reported in this work isn't particularly stronger than the provided baseline, and not many baselines are considered.
> A4: We apologize for the confusion. First, the term "Baseline" in our charts refers to the performance of models fine-tuned using ALL available data. Using a small amount of data can yield similar or even better performance, which aligns with the goal of data filtering. Second, for data filtering in SFT, we compared our method with the popular ChatGPT Selection method, also known as the AlpaGasus method mentioned in our related work. We achieved similar or better fine-tuning results using only one-third of the data with this method. Third, our approach is not about "excluding harmful data"; rather, we aim to identify data that the model can effectively learn during fine-tuning, without assuming a priori whether the data is "harmful" or "beneficial."
>
> Q5: It is unknown how to set the threshold in the proposed constraints.
> A5: Indeed, the method of selecting the threshold is missing from our current manuscript and we do intend to include it in the following revision. One possibility that we believe is pertinent to the current LLM research effort is to simply choose the threshold which results in a total number of data remaining for fine-tuning that can be supported by a given set of computational resources.
>
> Q6: Format: The appendix is not cut from the main paper.
> A6: Thank you for reviewing our appendix. However, ICLR has specific requirements for appendices: "Authors may use as many pages of appendices (after the bibliography) as they wish, but reviewers are not required to read the appendix." It appears there may be confusion with requirements from other conferences.

---

### Official Review · Reviewer_rNQa · 2023-11-02

**Soundness:** 2 fair
**Presentation:** 3 good
**Contribution:** 2 fair
**Rating:** 3
**Confidence:** 3

**Summary:**

The authors present a novel metric for example selection for
supervised fine tuning, inspired by a learnability principle.

**Strengths:**

Selecting fine tuning samples based on learnability principles seems a sensible idea.

The paper is easy to follow.

**Weaknesses:**

The novelty of the proposed learnability metric is unclear to me. The
authors propose three different criteria that are just different
aspects of the same criterion, namely relative loss reduction, which
is the normalized formula they eventually derived (albeit they
apparently did not recognize it as such).

The experimental evaluation is not entirely convincing:

About the comparison with ChatGPT selection: why not using the same
number of data points? e.g. 9,229 points also for your approach? the
comparison is not on equal grounds otherwise.

No comparison in made with alternative data selection procedures, even
if a number of them are listed in the related work section. I don't
think these can be dismissed without a comparison if the authors are
to claim that their approach is a general solution to SFT.

The robustness of the approach and the generality of the results
should be better assessed. For instance, Figure 4b indicates an
oscillatory behaviour that can be detrimental to the method. I believe
multiple datasets should be tested to present robust results.

**Questions:**

why not using the same number of data points in the comparison with ChatGPT select?

How did you enroll participants for human evaluation? how many did you have?

Also please comment on the concerns I raised in the weaknesses.

---

> ### Author Response · Authors · 2023-11-22
>
> We thank the reviewer for the comment. We plan on extensively revising our manuscript with more extensive benchmarking and will be withdrawing our submission as a result.
> However, we'd like to address a few points raised by the reviewer for clarification.
>
> Q1: The novelty of the proposed learnability metric is unclear.
> A1: We apologize for the confusion. For clarification, while the problem of training data selection is not new, its application to instruction-tuning datasets in LLMs has only received recent attention by related works contemporary to ours. Compared to these techniques, our method different itself by 1) taking into account the model capability rather than just perceived data quality, 2) avoiding the use of proprietary LLMs which may result in data leakage, 3) supporting an efficient data selection technique that could be used simultaneously with model training (as the losses for each SFT data for both pre-train and finetuned model can be calculated during training).
>
> Q2: Why not use the same number of data points? e.g. 9,229 points.
> A2: We apologize for the confusion. Indeed, we could utilize all 9,229 data points; however, in this experiment, we intend to demonstrate that a smaller training set selected using our proposed method (with only around 3,000 samples) can achieve superior results to using the full 9.2K data points. In the next revision, we plan to compare the performance of the model fine-tuned with our method using the full 9,229 data points against the method employing ChatGPT Select. This comparative analysis aims to enhance the rigor of our experimental design.
>
> Q3：No comparison is made with alternative data selection procedures.
> A3: We apologize for any confusion. In this context, the "ChatGPT Selection" method refers to the AlpaGasus method that we referenced in the paper. The experimental results indicate that our approach significantly outperforms AlpaGasus. We conducted an extensive review of recent relevant literature, and AlpaGasus emerged as one of the most effective methods, serving as our chosen baseline. In the "Related Works" section, we also mentioned the Humpback study. However, due to its relatively complex methodology and process, which does not align with a general SFT data selection approach, we did not include it in our comparative analysis.
>
> Multiple datasets should be tested to present robust results.
> A4: Thank you very much for your suggestions. Our approach does not rely on the Alpaca dataset; we use it simply because it is one of the most widely used SFT datasets currently available. The introduction of Alpaca-3.5 and Alpaca-4 datasets in the paper is aimed at demonstrating the generality of our method on both high and low-quality datasets. We plan to incorporate additional datasets into future experiments to enrich our research further. Once again, we appreciate your suggestions.
>
> Q5: How did you enroll participants for human evaluation? how many did you have?
> A5: We apologize for not mentioning this in the paper due to space constraints, as most similar works also do not delve into these details. We engaged three researchers who were not involved in the project to address this limitation. They were presented with prompts in a random order, and unaware of which model generated the responses. Their task was to choose between options such as "A is better than B," "A is worse than B," or "A and B are similar," and the majority choice was recorded as the result. We believe this setup is reasonable and avoids introducing biases from the project's researchers, providing a relatively objective reflection of human preferences. The limitation lies in the relatively small number of participants, and we acknowledge the potential to hire more annotators for future work.